# A Multiscale Frequency Domain Causal Framework for Enhanced Pathological Analysis

**Xiaoyu Cui**[1*]   **Weixing Chen**[2*]   **Jiandong Su**[3*]
[1]Northeastern University    [2] Sun Yat-sen University    [3]Shenzhen Institute of Advanced Technology
`cuixy@bmie.neu.edu.cn, chenwx228@mail2.sysu.edu.cn,`
`jiandong.laurence.su@gmail.com`

## Abstract

Multiple Instance Learning (MIL) in digital pathology Whole Slide Image (WSI) analysis has shown significant progress. However, due to data bias and unobservable confounders, this paradigm still faces challenges in terms of performance and interpretability. Existing MIL methods might identify patches that do not have true diagnostic significance, leading to false correlations, and experience difficulties in integrating multi-scale features and handling unobservable confounders. To address these issues, we propose a new Multi-Scale Frequency Domain Causal framework (MFC). This framework employs an adaptive memory module to estimate the overall data distribution through multi-scale frequency-domain information during training and simulates causal interventions based on this distribution to mitigate confounders in pathological diagnosis tasks. The framework integrates the Multi-scale Spatial Representation Module (MSRM), Frequency Domain Structure Representation Module (FSRM), and Causal Memory Intervention Module (CMIM) to enhance the model's performance and interpretability. Furthermore, the plug-and-play nature of this framework allows it to be broadly applied across various models. Experimental results on Camelyon16 and TCGA-NSCLC dataset show that, compared to previous work, our method has significantly improved accuracy and generalization ability, providing a new theoretical perspective for medical image analysis and potentially advancing the field further. The code will be released at `https://github.com/WissingChen/MFC-MIL`.

## 1 Introduction

The classification of Whole Slide Images (WSIs) involves the use of automated techniques to extract critical features from pathological slides and perform classification, thereby aiding in disease diagnosis Litjens et al. (2017). This technology has the potential to enhance diagnostic efficiency and accuracy while reducing human error, which is particularly crucial for the early detection of cancer Madabhushi & Lee (2016). However, this task is challenging due to the high resolution of the images, which can contain billions of pixels, with diagnostically relevant regions often comprising only a small portion. This complexity complicates the process of feature identification Komura & Ishikawa (2018). Furthermore, variations in staining techniques and other sources of noise in the images can introduce model bias and lead to misjudgment Srinidhi et al. (2021); Li et al. (2024).

To address these challenges, multiple instance learning (MIL) methods have been developed for the classification of WSIs. MIL methods reduce the computational requirements of WSI by selecting multiple diagnostically relevant patches to represent the entire image, demonstrated in extensive tasks Ilse et al. (2018); Shao et al. (2021); Yao et al. (2019; 2020); Xu et al. (2019). Moreover, strategies for selecting these relevant patches help minimize the impact of noise, thereby improving the model's effectiveness Zheng et al. (2024). However, the inclusion of redundant, irrelevant information can introduce data biases, potentially causing these methods to mistakenly associate non-causal

---

*Equal Contribution.

features with diagnostic outcomes as confounders, leading to erroneous conclusions Schwab et al. (2019); Chen et al. (2023).

Spurious correlations are caused by confounders that, while co-occurring with disease states in the data, do not have a direct causal relationship with disease outcomes Pearl & Mackenzie (2018); Liu et al. (2022b). As illustrated in Figure 1 (a), if positive and negative samples in the training set are predominantly associated with specific colors, the model may incorrectly associate color with pathological categories, leading to spurious correlations. Consequently, when positive samples in the test set share the same color as negative samples, the model may misclassify them due to these previously established spurious correlations. The Structural Causal Model (SCM) analysis Pearl et al. (2016) suggests that this error occurs because the model fails to correctly follow the causal link $X \to Y$, instead relying on an incorrect causal path, $X \leftarrow Z \to Y$, as shown in Figure 1 (b).

To eliminate these spurious correlations, IBMIL Lin et al. (2023) utilize causal back-door interventions by estimating and removing confounders $Z$ to reduce bias, as shown in Figure 1 (c). However, the two-step training strategy used in IBMIL increases both the complexity and computational cost of the methods. Similarly, CaMIL Chen et al. (2024) adopt the front-door intervention for deconfounding with the estimation of mediator $M$ as shown in Figure 1 (d). However, CaMIL relies on preprocessed features to represent the overall distribution of the dataset and requires time-consuming feature clustering processes. Moreover, through in-depth analysis of pathological diagnosis, we find that tissue structure at low magnification and cellular structure at high magnification is critical for accurate diagnosis Schmitz et al. (2021). Existing methods generally handle features from a single magnification or rely on preprocessed multi-magnification features, which not only increase computational complexity but also hinder the comprehensive capture of spatial relationships across multiple levels Li et al. (2021). Additionally, current structural information extraction methods are often vulnerable to interference from image staining techniques and color contrast, leading to misclassifications Vahadane et al. (2016); Tellez et al. (2019).

To address these challenges, we propose the Multi-Scale Frequency Domain Causal (MFC) framework, which consists of three key components: the Causal Memory Intervention Module (CMIM), the Multiscale Spatial Representation Module (MSRM), and the Frequency-domain Structural Representation Module (FSRM), as illustrated in Figure 1 (e). The CMIM is designed to mitigate data bias by preventing the model from relying on spurious correlations for decision-making. By preserving critical diagnostic features as learnable memory features, CMIM facilitates plug-and-play causal interventions, eliminating unobservable confounders' misleading effects. The MSRM addresses the challenge of integrating multilevel information by combining positional encoding with multiscale large-kernel convolutions, enabling the model to capture the spatial relationships between low-magnification tissue structures and high-magnification cellular structures, thereby enhancing its ability to represent multilevel features. Finally, the FSRM integrates phase information in the frequency domain to reduce interference from staining techniques and color contrast, extracting structural information that is directly related to diagnostic outcomes. Together, these three modules enable the MFC framework to perform pathology image classification tasks with greater accuracy and robustness.

## 2 RELATED WORK

### 2.1 MULTIPLE INSTANCE LEARNING

In the task of WSI classification, Multiple Instance Learning (MIL) methods have become the predominant approach for handling high-resolution pathology images by segmenting them into multiple patches and aggregating these patches at the bag level to achieve efficient classification Ilse et al. (2018); Li et al. (2021); Shao et al. (2021); Zhang et al. (2022). These methods utilize various aggregation strategies and attention mechanisms to enhance the model's ability to capture critical information, thereby improving classification performance. However, they still face limitations in accurately selecting instances and recognizing complex pathological features, particularly when dealing with heterogeneous data, which can lead to information loss or misclassification. To address these issues, researchers have proposed several enhancements, such as optimizing instance selection to improve the efficiency of utilizing critical patches Lu et al. (2021); Zheng et al. (2024), applying stain normalization techniques to reduce color variation between samples Tellez et al. (2019), and enhancing model adaptability across different datasets through domain generalization Stacke et al.

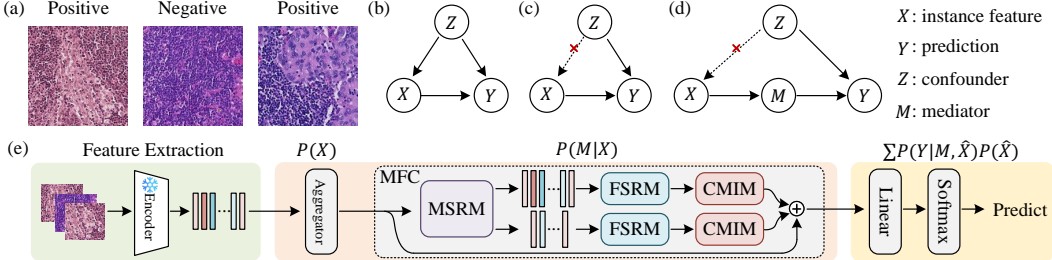

Figure 1: (a) WSI sample. (b-d) Causal diagrams for no intervention, back-door intervention, and front-door intervention. (e) We propose Multi-Scale Frequency Domain Causal Multi-Instance Learning (MFC-MIL), a framework that is plug-and-play compatible with various MIL models. In the MIL model, WSI is typically encoded as CLS tokens and feature tokens (patch tokens). Our MFC framework extends this approach by applying the MSRM and FSRM modules specifically to the feature tokens. The output from these modules is then fed into the memory module of the CMIM for interaction, ultimately influencing the CLS token.

(2020). While these improvements often increase robustness and generalization, they also tend to add computational complexity and still fall short of effectively integrating multilevel pathological information Li et al. (2021). In contrast, our proposed approach introduces multilevel spatial representation and causal intervention mechanisms, which not only simplify the model architecture but also effectively address challenges arising from spurious correlations and complex feature integration, significantly improving both accuracy and generalization capabilities.

## 2.2 CAUSAL INFERENCE

In medical image analysis, causal intervention methods aim to explicitly model and intervene in causal relationships to reduce the model's reliance on spurious correlations, thereby improving diagnostic accuracy Castro et al. (2020); Liu et al. (2023). These approaches typically involve identifying and adjusting for potential confounders Nie et al. (2023); Lin et al. (2022) or mediators Chen et al. (2023) to correct spurious correlations caused by bias during inference, leading to a more accurate capture of causal relationships relevant to disease outcomes. In WSI classification, IBMIL Lin et al. (2023) and CaMIL Chen et al. (2024) are two prominent causal intervention strategies. IBMIL employs a back-door intervention to eliminate the influence of confounders on classification results, but its two-stage training process is complex and requires the prior identification and maintenance of a confounder set, increasing implementation difficulty. CaMIL, by contrast, uses a front-door intervention strategy to remove unobservable confounders via mediators. However, its reliance on clustering methods to select mediators during training increases computational time and may reduce the interpretability of the mediators. In contrast, our approach combines multilevel spatial representation with frequency-domain structural representation as mediators, simplifying the training process by avoiding cumbersome feature clustering and complex confounder management. This not only enhances computational efficiency but also improves the model's causal interpretability and classification accuracy.

## 3 METHOD

In this section, we first introduce the Causal Memory Intervention Module (CMIM), followed by a discussion of two key representation modules: the Multiscale Spatial Representation Module (MSRM) and the Frequency-domain Structural Representation Module (FSRM). We then explain how MSRM and FSRM are integrated into the MIL framework and how CMIM is utilized for causal intervention.

### 3.1 CAUSAL MEMORY INTERVENTION MODULE

MIL models for WSI classification are tasked with detecting the presence of positive instances within a bag containing numerous instances. If at least one positive instance is present, the bag is classified as positive; otherwise, it is classified as negative. We assume that $S =$

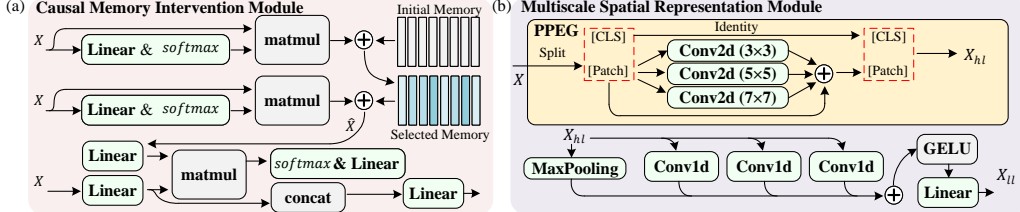

Figure 2: The illustration of our proposed modules in MFC-MIL, including the Causal Memory Intervention Module and Multiscale Spatial Representation Module.

$(p_1, y_1), (p_2, y_2), \ldots, (p_n, y_n)$ represents a WSI sample $S$ containing $n$ patches $p$, with corresponding instance labels $y$. The bag label $Y$ can then be formulated as follows:

$$Y = \begin{cases} 0, & \text{iff } \sum y_i = 0 \\ 1, & \text{otherwise} \end{cases} \tag{1}$$

Since the instance-level labels are usually unavailable, we must estimate the bag-level label based on the predicted instance-level labels. Specifically, in the MIL framework, the process involves using a frozen visual encoder $f(\cdot)$ to map patch-level images into low-dimensional feature representations, which constitutes the feature extraction step. An aggregation module $\theta_a(\cdot)$ is then employed to combine these instance-level features into a bag-level representation, followed by a classification module $\theta_c(\cdot)$ to estimate the bag-level label $Y$, as formulated as:

$$Y = \theta_c(\theta_a(x_1, x_2, ..., x_n)), x_i = f(p_i). \tag{2}$$

However, due to the presence of confounders, the non-causal model does not follow the correct causal path $X \rightarrow Y$ for prediction. Instead, it relies on spurious correlations established by the confounders, following the incorrect path $X \leftarrow Z \rightarrow Y$, which can be formulated as:

$$P(Y|X) = \sum_z P(Y|X, Z = z)P(Z = z|X), \tag{3}$$

where, $X = \{x_1, x_2, ..., x_n\}$ represents the patch-level features and $Z$ represents the confounders that lead to spurious correlations, which are typically difficult to estimate, especially in the absence of a well-trained semantic extractor. Therefore, the features generated by the aggregator $\theta_a$ are treated as the mediators, and the do-operator $do(\cdot)$ is introduced to apply causal front-door intervention for deconfounding, effectively cutting off the link $X \leftarrow Z \rightarrow Y$. The total probability $P(Y|do(X))$ can be expressed as the following summation:

$$P(Y|do(X)) = \sum_m P(Y|do(X), M = m)P(M = m|do(X)), \tag{4}$$

where, $M$ is introduced by $X$ without any back-door path, and there is no direct causal relationship between $X$ and $Y$. Additionally, the link $M \leftarrow X \leftarrow Z \rightarrow Y$ can be further severed to achieve deconfounding. Finally, the Eq. 4 can be rewritten as follows:

$$P(Y|do(X)) = \sum_m P(M = m|X = x) \sum_{\hat{x}} P(X = \hat{x})P(Y|X = \hat{x}, M = m). \tag{5}$$

where $\hat{x}$ represents the potential estimated values selected from $x$. The detailed derivation can be found in the Appendix A.1. However, previous methods typically use clustering to estimate $\hat{x}$, which significantly hinders computational efficiency. Therefore, we propose utilizing a memory module to estimate the overall distribution of the dataset during training and to refine the estimation of $\hat{x}$ through attention-based sampling. Specifically, a set of trainable parameters with length $k$ is initialized as memory and combined with attention-weighted inputs to select relevant memory elements. This selected memory is then further sampled and used as $\hat{x}$ in the front-door intervention,

as illustrated in Figure 2 (a), (more detail can be found in Appendix A.1). Finally, we employ the Normalized Weighted Geometric Mean (NWGM) Liu et al. (2022a) to estimate the equation.

Specifically, to implement this equation effectively, we need to address two challenges: (1) estimating the overall distribution $P(X = \hat{x})$ without explicit clustering, and (2) ensuring that the mediator $M$ captures causal structural information while being robust against confounders. We solve the first challenge through our memory module, which learns to represent the distribution of features across the dataset. For the second challenge, we employ specialized frequency-domain and multiscale representations that focus on structural patterns rather than color or staining characteristics that often serve as confounders in pathological images. This approach allows us to implement the front-door adjustment in an end-to-end trainable framework without requiring explicit identification of confounders or mediators.

## 3.2 MULTISCALE SPATIAL REPRESENTATION MODULE

After implementing CMIM, we further refined the estimation of the mediator, particularly by integrating low-magnification tissue information with high-magnification cellular information, which are both crucial in the diagnostic process. We propose the Multiscale Spatial Representation Module (MSRM), which first applies Position-aware Patch Embedding Generation (PPEG) Shao et al. (2021) for multiscale positional encoding, followed by sampling with multiple large-kernel convolutions, as illustrated in Figure 2 (b).

Specifically, $X \in R^{\{N,D\}}$ is padded using the sampled patches in the PPEG and reshaped to $X \in R^{\{D, \frac{N_p}{2}, \frac{N_p}{2}\}}$, where $N$ represents the number of patches in the current slide, $D$ is the model dimension, and $N_p$ is the number of patches after padding. After passing through 2D convolutional layers with kernel sizes of 7, 3 and 5, the padding is removed, and the original dimensions are restored, resulting in high-resolution features $X_{hl} \in R^{\{N,D\}}$. Then, three 1D convolutional layers, each with a kernel size of 16, are applied with dilation rates of 1, 3 and 5 to extract features with multiscale receptive fields. This process can be formalized as follows:

$$X_{ll} = \text{Linear}(\text{Conv}_1(X_{hl}) + \text{Conv}_2(X_{hl}) + \text{Conv}_3(X_{hl}) + \text{MaxPooling}(X_{hl})) \quad (6)$$

Here, $X_{ll} \in R^{\{\frac{N}{16}, D\}}$ represents the low-magnification information obtained through sampling, while the MaxPooling layer is used as a residual connection to provide additional information about tissue contours. Additionally, the joint dimension between the convolutional layer and linear is $D_j$

## 3.3 FREQUENCY-DOMAIN STRUCTURAL REPRESENTATION MODULE

The Hilbert transform is a tool utilized to derive the analytic representation of a real-valued signal. Applying the Hilbert transform to a signal $x(t)$, we obtain a complex-valued function $x_a(t)$, where the original signal forms the real part, and the Hilbert transform provides the imaginary part. The analytic signal $x_a(t)$ can be expressed as:

$$x_a(t) = x(t) + j\hat{x}(t) \quad (7)$$

where $\hat{x}(t)$ is the Hilbert transform of $x(t)$ given by:

$$\hat{x}(t) = \frac{1}{\pi}\text{P.V.} \int_{-\infty}^{\infty} \frac{x(\tau)}{t - \tau} \, d\tau \quad (8)$$

P.V. denotes the Cauchy principal value of the integral.

In this complex representation, the real part $x(t)$ retains the original amplitude information of the signal, representing its observable component. The imaginary part $\hat{x}(t)$, on the other hand, provides the quadrature component, which is essential to understand the phase characteristics of the signal. Importantly, the imaginary part $\hat{x}(t)$ is the phase-shifted version 90 degrees of the real part $x(t)$. By combining the real and imaginary parts, we can obtain a comprehensive representation that captures significantly richer information than spatial domain features alone.

The Frequency-domain Structural Representation Module (FSRM) addresses the need to extract nuanced and informative features from Whole Slide Images (WSIs) to enhance classification accuracy.

Traditional spatial-domain methods often fail to capture subtle pathological indicators, risking the loss of critical diagnostic information. Using frequency domain analysis, the FSRM reveals hidden patterns and relationships within image data that may not be evident in the spatial representation. This capability is particularly advantageous in WSI analysis, where complex tissue structures and cellular arrangements exhibit distinctive frequency signatures.

The FSRM consists of several key components working in concert. First, an input projection layer maps initial features to an appropriate dimensional space, preparing them for frequency-domain transformation. At the module's core, the Hilbert transform is implemented through Fast Fourier Transform (FFT), which first calculates the FFT of the signal, then constructs a frequency domain filter (doubling the positive frequency components), and finally obtains the real part through inverse FFT. The transformed features are subsequently mapped back to the original dimensional space via an output projection layer. To maintain original feature information and support gradient flow during training, the entire module integrates a residual connection.

The overall transformation function $F : \mathbb{R}^d \to \mathbb{R}^d$ is defined as:

$$F(\mathbf{x}) = \mathbf{x} + g(H(f(\mathbf{x}))) \tag{9}$$

where: $f : \mathbb{R}^d \to \mathbb{R}^{512}$ is the input linear mapping $f(\mathbf{x}) = \mathbf{W}_1\mathbf{x} + \mathbf{b}_1$ and $g : \mathbb{R}^{512} \to \mathbb{R}^d$ is the output linear mapping $g(\mathbf{x}) = \mathbf{W}_2\mathbf{x} + \mathbf{b}_2$. $H$ is the Hilbert transform operator. The final output can be expressed as: $\mathbf{y} = \mathbf{x} + g(H(f(\mathbf{x}))) = \mathbf{x} + \mathbf{W}_2(H(\mathbf{W}_1\mathbf{x} + \mathbf{b}_1)) + \mathbf{b}_2$. This design allows the FSRM to enhance feature representation by capturing complex structural and textural variations critical for robust WSI classification. It is worth emphasizing that the FSRM and CMIM modules play complementary roles within our framework. While the Hilbert Transform focuses on extracting frequency-domain structural information, endowing the representation with robustness against superficial artifacts such as staining bias, the CMIM performs causal intervention to eliminate confounding factors within the data.

## 4 EXPERIMENT

### 4.1 DATASET AND METRIC

The Camelyon16 dataset Bejnordi et al. (2017) is widely used for detecting breast cancer metastases. It comprises 270 training and 129 testing images, segmented into about 3.2 million patches of $256 \times 256$ pixels at 20× magnification, averaging 8,000 patches per slide. Meanwhile, the TCGA-NSCLC dataset focuses on two lung cancer subtypes, LUSC and LUAD, with 1,054 whole slide images. It is divided into training, validation, and test sets in a 7:1:2 ratio, yielding 5.2 million patches at 20× magnification, with about 5,000 patches per slide.

To evaluate the effectiveness of our approach, we apply four key metrics for classification performance: accuracy, F1 score, specificity, and the area under the receiver operating characteristic curve (AUC). These metrics provide a comprehensive assessment of the method's overall performance.

### 4.2 IMPLEMENTATION SETTINGS

In the feature extraction process, we employed a CNN-based ResNet18, with parameters pre-trained using SimCLR as part of the DSMIL framework. The model operates with a dimension of 512, while the value of $k$ in CMIM is set to 16 for high-resolution features and 32 for low-resolution features. For most experiments, we used the Adam optimizer with an initial learning rate of 2e-4 and a weight decay of 5e-4. Additionally, our MFC estimates the mediator using patch-level features and applies it to intervene in the aggregated bag-level prediction vector. The mini-batch size used for training is 1, and the model is trained for 100 epochs. All experiments were conducted on an NVIDIA GeForce RTX 2080Ti.

### 4.3 BASELINE

**ABMIL** Ilse et al. (2018) enhances multi-instance learning through attention mechanisms, focusing on critical image regions to improve pathological classification performance. **DSMIL** Li et al. (2021) employs a dual-stream network structure with self-supervised contrastive learning to enhance the accuracy of whole-slide image classification. **TransMIL** Shao et al. (2021) utilizes a

transformer-based approach for relevant multi-instance learning, aiming to better capture key information within images. **CLAM** Chen et al. (2024) leverages clustering-constrained attention-based multiple instance learning to enable efficient, interpretable, and adaptable slide-level pathology classification without manual annotations. **DTFD-MIL** Zhang et al. (2022) leverages a dual-layer feature distillation strategy for multi-instance learning, optimizing the performance of tissue pathology classification for whole-slide images.

## 4.4 EXPERIMENT RESULT

Table 1 presents the results of our WSI classification experiments on two benchmark datasets, Camelyon16 and TCGA-NSCLC, using 5-fold cross-validation. Overall, all MIL methods showed significant improvement after applying the MFC framework, demonstrating the effectiveness of our approach. Specifically, the DSMIL method, which models multiscale features, achieved an average accuracy gain of 5.27% on Camelyon16 and 2.08% on TCGA-NSCLC. This indicates that the proposed MFC framework can more effectively leverage multiscale information. Moreover, we observed that the performance improvements on the Camelyon16 dataset were generally greater than those on the TCGA-NSCLC dataset, consistent with the findings from IBMIL Lin et al. (2022). This can be attributed to the more severe data bias in the Camelyon16 dataset, where positive bags contain only a small fraction of positive instances (approximately less than 10%). This difference further highlights the ability of MFC to effectively identify multiscale structural information (the mediators), that is relevant to diagnosis and to intervene in bag-level predictions.

Additionally, it can be observed that our MFC framework significantly improves accuracy and the F1 metric, while the enhancement in the AUC metric is relatively minor. Notably, in methods such as CLAM-SB and CLAM-MB on the Camelyon16 dataset, the performance is worse compared to the baseline in terms of AUC, despite substantial improvements in other metrics. This suggests that MFC alters the sample distribution in the data, such that the model better handles certain boundary samples, leading to increases in F1 and specificity. However, the handling of non-boundary samples is less balanced than before, which could negatively impact the overall AUC performance. This effect is particularly pronounced in high-dimensional, complex data such as pathological image classification, where the treatment of boundary and misclassified samples may significantly affect specific metrics. Since AUC provides a more comprehensive evaluation, it is likely more sensitive to these subtle changes, especially when the baseline model already performs well on this metric.

Furthermore, we reproduced the IBMIL model and conducted a five-fold cross-validation using DSMIL as the baseline on the Camelyon16 dataset. As shown in the table, our method consistently outperforms IBMIL, even though IBMIL also surpasses the baseline's performance. However, as previously mentioned, while MFC effectively handles certain boundary samples and further improves accuracy and the F1 metric, it also results in a suboptimal performance in the AUC metric.

## 4.5 ABLATION STUDIES

To further evaluate the effectiveness of our method, we conducted detailed and comprehensive ablation experiments, including the removal of individual modules and tests on key parameter settings for each module.

### 4.5.1 EFFECTIVENESS OF CMIM

As shown in Table 3, the CMIM model significantly outperforms the baseline, particularly exhibiting an improvement of nearly 10% in the specificity metric. This suggests that CMIM is more effective in capturing causal features, especially in distinguishing negative samples. However, the enhanced performance on negative samples shifts the decision boundary, making the identification of positive samples more conservative, which in turn leads to a decrease in recall.

To further investigate the impact of the memory mechanism in CMIM, we conducted an experimental analysis of the number of memory slots, denoted as $k$, within the MFC-MIL framework. Specifically, we input cellular structural features ($x_{hl}$) and tissue structural features ($x_{ll}$) into the memory module, utilizing the activated memory for front-door intervention. In these experiments, we fixed the memory slot count for $x_{ll}$ at 32 and varied the memory slots for $x_{hl}$ at 4, 8, 16, 32, and 48. As shown in Figure 3 (a), the model's performance improved steadily when $k$ ranged from

| | Camelyon16 | | | | | TCGA-NSCLC | | | |
|---|---|---|---|---|---|---|---|---|---|
| Method | ACC | AUC | F1 | Spe. | Method | ACC | AUC | F1 | Spe. |
| ABMIL | 88.84(2.49) | 95.65(1.85) | 84.40(4.16) | 93.75(7.81) | ABMIL | 91.38(1.88) | 99.10(0.06) | 90.72(2.3) | 98.00(1.17) |
| + MFC | 91.78(1.87) | 97.68(0.29) | 88.94(2.07) | 95.00(4.76) | + MFC | 92.23(1.03) | 99.24(0.26) | 91.70(1.24) | 98.46(0.54) |
| Δ | +2.94(3.11) | +2.03(1.87) | +4.54(4.65) | +1.25(9.15) | Δ | +0.85(2.14) | +0.14(0.27) | +0.98(2.61) | +0.46(1.29) |
| DSMIL | 86.98(4.73) | 94.95(4.43) | 79.41(9.11) | 98.00(2.88) | DSMIL | 89.61(1.72) | 96.75(0.13) | 88.61(2.02) | 98.46(0.34) |
| + MFC | 92.25(2.33) | 95.41(1.12) | 89.13(3.42) | 97.25(1.63) | + MFC | 91.69(1.04) | 98.95(0.13) | 91.20(1.31) | 98.08(1.84) |
| Δ | +5.27(5.27) | +0.46(4.57) | +9.72(9.73) | -0.75(3.31) | Δ | +2.08(2.01) | +2.2(0.18) | +2.59(2.41) | -0.38(1.87) |
| TransMIL | 84.50(2.74) | 94.88(1.73) | 80.90(1.58) | 83.50(10.25) | TransMIL | 91.54(2.39) | 98.44(0.52) | 90.93(2.87) | 97.38(2.01) |
| + MFC | 90.85(1.18) | 97.68(1.07) | 88.00(1.13) | 92.75(5.77) | + MFC | 92.85(0.97) | 98.98(0.12) | 92.50(1.13) | 97.53(0.69) |
| Δ | +6.35(2.98) | +2.80(2.03) | +7.10(1.94) | +9.25(11.76) | Δ | +1.31(3.09) | +0.54(0.53) | +1.07(3.08) | +0.15(2.13) |
| CLAM-SB | 86.67(7.09) | 96.89(1.44) | 84.30(5.51) | 84.75(16.04) | CLAM-SB | 90.85(1.60) | 99.05(0.12) | 90.12(1.95) | 97.85(0.84) |
| + MFC | 89.77(1.93) | 96.11(1.24) | 85.87(2.10) | 88.50(3.99) | + MFC | 91.31(0.89) | 99.22(0.16) | 90.67(1.07) | 98.00(0.69) |
| Δ | +2.01(7.35) | -0.69(1.90) | +1.57(5.90) | +3.75(16.53) | Δ | +0.46(1.83) | +0.17(0.2) | +0.55(2.22) | +0.15(1.09) |
| CLAM-MB | 88.99(2.65) | 97.65(0.32) | 85.18(3.82) | 82.00(9.91) | CLAM-MB | 91.22(2.91) | 99.01(0.08) | 90.85(3.56) | 97.99(0.54) |
| + MFC | 91.94(0.55) | 97.29(0.52) | 88.44(0.75) | 98.50(1.63) | + MFC | 91.85(1.42) | 99.10(0.11) | 91.40(1.91) | 98.46(4.35) |
| Δ | +2.95(2.71) | -0.36(0.61) | +3.26(3.89) | +16.5(10.04) | Δ | +0.63(3.24) | +0.09(0.14) | +0.55(4.04) | +0.47(4.38) |
| DTFD (MaxS) | 85.89(13.40) | 97.59(0.07) | 71.41(39.94) | 95.00(4.05) | DTFD (MaxS) | 81.31(0.75) | 88.52(0.04) | 80.32(0.58) | 86.31(2.95) |
| + MFC | 92.09(1.77) | 97.65(0.22) | 88.62(2.89) | 98.50(2.71) | + MFC | 91.23(1.66) | 98.88(0.07) | 90.54(2.06) | 98.15(0.88) |
| Δ | +6.2(13.52) | +0.06(0.23) | +17.21(40.04) | +3.5(4.87) | Δ | +9.92(1.82) | +10.36(0.08) | +10.22(2.14 ) | +11.84(3.08) |

Table 1: Main result (%) on Camelyon16 and TCGA-NSCLC, the value in parentheses is the standard deviation of the 5-fold cross-validation. However, the values in parentheses in the $\Delta$ column represent the propagated uncertainties, denoted as $\sigma\Delta = \sqrt{(\sigma A^2 + \sigma B^2)}$, where $A$ corresponds to the base model and $B$ refers to the model enhanced using MFC.

| Method | ACC | AUC | Pre. | Rec. | F1 | Spe. |
|---|---|---|---|---|---|---|
| Baseline | 86.98 | 94.95 | 96.14 | 68.98 | 79.41 | 98.00 |
| IBMIL | 91.78 | **96.31** | 94.85 | 83.67 | 88.50 | 96.75 |
| MFC-MIL | **92.25** | 95.41 | **94.96** | **84.08** | **89.13** | 97.25 |

Table 2: Comparison of the results (%) of our MFC-MIL and IBMIL on the Camelyon16 dataset, and the baseline is DSMIL.

| CMIM | MSRM | FSRM | ACC | AUC | Pre. | Rec. | F1 | Spe. |
|---|---|---|---|---|---|---|---|---|
| | | | 84.50 | 94.88 | 78.01 | 86.12 | 80.90 | 83.50 |
| √ | | | 88.37 | 97.53 | 89.92 | 81.63 | 83.78 | 92.50 |
| √ | √ | | 89.46 | 97.61 | **91.61** | 81.63 | 85.45 | **94.25** |
| √ | √ | √ | **90.85** | **97.68** | 89.25 | **87.76** | **88.00** | 92.75 |

Table 3: Ablation result (%) of MFC-MIL on Camelyon16 dataset, and the baseline model is TransMIL.

4 to 16, suggesting that increasing the number of memory slots enhances the model's ability to capture cell-level pathological features. However, when $k$ increased to 32 or 48, performance declined, implying that an excessive number of memory slots may introduce redundant information. In high-dimensional feature spaces, this could result in capturing noise or irrelevant features, leading to overfitting and reduced generalization capacity.

Furthermore, in experiments where $x_{hl}$ was fixed at 16 memory slots while varying the memory slot count for $x_{ll}$ (Figure 3 (b)), a similar trend was observed. Both ACC and AUC improved as $k$ increased from 4 to 16 but declined when $k$ reached 48. Interestingly, the F1 and specificity metrics followed an opposite pattern. This may be due to $x_{ll}$ representing global tissue-level structural information, which emphasizes macroscopic pathological structures. At lower magnifications, the model better distinguishes normal tissues, resulting in a significant increase in specificity. However, as the reliance on global information grows, the model's sensitivity to subtle cell-level pathological features diminishes, leading to more false negatives. This explains why, at higher $k$ values, F1 scores decrease while specificity remains high, approaching 100%.

### 4.5.2 EFFECTIVENESS OF MSRM

As shown in Table 3, the MSRM module effectively captures information across different scales, from local details to global structures, by processing features at multiple scales. This multi-scale feature integration significantly improves the model's performance, particularly in terms of specificity and precision. The high specificity indicates the model's enhanced ability to identify negative

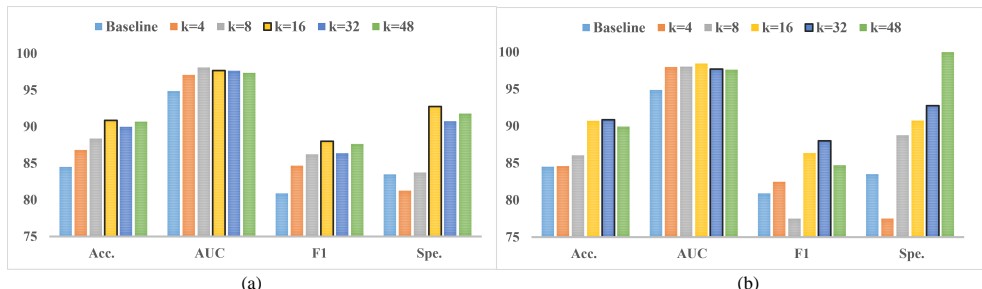

Figure 3: Ablation studies of the memory number $k$ of $x_{hl}$ (shown as (a)) and $x_{ll}$ (shown as (b)) for CMIM on Camelyon16 dataset, and the baseline is TransMIL.

| Joint Dimension of MSRM | | | | | Frequency Information Extraction of FSRM | | | | |
|---|---|---|---|---|---|---|---|---|---|
| size of $D_j$ | ACC | AUC | F1 | Spe. | Method | ACC | AUC | F1 | Spe. |
| Baseline | 84.50 | 94.88 | 80.90 | 83.50 | Baseline | 84.50 | 94.88 | 80.90 | 83.50 |
| 512 | **90.85** | 97.68 | **88.00** | 92.75 | FFT | 88.37 | 91.66 | 83.52 | 95.00 |
| 1024 | 89.92 | **97.88** | 87.62 | 87.5 | DCT | 89.15 | 92.19 | 84.44 | 96.25 |
| 2048 | 87.6 | 97.3 | 80.49 | **100** | DWT | 89.15 | **97.93** | 83.72 | **98.75** |
| 4096 | 84.5 | 97.73 | 82.46 | 77.5 | Hilbert transform | **90.85** | 97.68 | **88.00** | 92.75 |

Table 4: Ablation result of MSRM and FSRM on Camelyon16 dataset, and the baseline is TransMIL.

samples. However, the unchanged Rec. suggests that while the model improves in recognizing negative samples, it does not yield further gains in detecting positive samples.

Additionally, as indicated in Table 4, experiments conducted with various $D_j$ settings demonstrate that the model achieves optimal performance on ACC and F1 metrics when $D_j$ is set to 512. This suggests that the original feature dimensionality is sufficient to capture relevant information, and further expansion is unnecessary. Although increasing the dimensionality to 1024, 2048, and 4096 theoretically enhances the model's expressive capacity, in practice, it leads to performance degradation. This decline may result from the model capturing excessive noise or irrelevant information, which negatively impacts generalization. Particularly at $D_j = 4096$, the model's performance becomes comparable to the baseline, indicating that excessive feature dimensionality expansion may completely offset the potential benefits of multi-scale feature extraction.

### 4.5.3 EFFECTIVENESS OF FSRM

In the FSRM, applying the Hilbert transform to both high-magnification and low-magnification features allows the model to capture richer structural information. In WSIs, frequency domain data reveals intricate details that are often missed in the traditional spatial domain. As shown in Table 3, the introduction of FSRM improved key metrics such as ACC, AUC, recall, and F1, demonstrating that this module significantly enhances the model's generalization ability and sensitivity to lesion areas. However, despite FSRM's strong performance in capturing structural features, there was a decline in precision and specificity compared to using only CMIM and MSRM. Specifically, precision dropped from 91.61% to 89.25%, and specificity decreased from 94.25% to 92.75%. This suggests that while FSRM improves the model's sensitivity to positive samples, it may reduce its ability to distinguish negative ones.

Analysis highlights the superiority of the Hilbert transform over the Fast Fourier Transform (FFT), Discrete Cosine Transform (DCT), and Discrete Wavelet Transform (DWT), particularly in AUC and F1 scores. The Hilbert transform's ability to capture instantaneous phase information is crucial for processing complex pathological images. In contrast, FFT and DCT focus on magnitude variations, often missing phase characteristics essential for detecting subtle changes. For example, FFT achieved an AUC of 91.66%, far below the 97.68% of the Hilbert transform, showing FFT's global averaging fails to represent local structural details effectively. DCT, while achieving high specificity (96.25%) by emphasizing low-frequency information, underperformed in recognizing positive samples. Despite this, DCT lagged behind the Hilbert transform in F1 and AUC, demonstrating the Hilbert transform better balances global and local features. Compared to DWT, which offers multi-resolution analysis but is limited by fixed basis functions, the Hilbert transform more effectively captures rapid intensity variations and local irregularities like cell membranes. Its robustness in ex-

tracting diagnostic features while resisting staining biases ensures higher precision in tasks requiring detailed structural analysis.

## 5 CONCLUSION

This study introduces MFC-MIL, a novel and flexible framework that addresses critical challenges in pathological image classification by leveraging multiscale spatial and frequency domain information. Through its three key modules—MSRM, FSRM, and CMIM—MFC-MIL not only enhances diagnostic accuracy but also demonstrates a deep understanding of the inherent complexities in WSIs. The model's ability to preserve spatial correlations across magnifications, mitigate color contrast variations, and reduce confounding factors reflects a sophisticated approach to medical image analysis. Future work could explore incorporating regularization methods based on Rényi entropy, as suggested in recent studies, to enhance feature representation and memory capacity Guan et al. (2024); Wang et al. (2024). Experimental results on the Camelyon16 and TCGA-NSCLC datasets highlight significant improvements in accuracy, F1 score, and specificity, underscoring the framework's ability to more precisely distinguish between positive and negative samples. This study offers important insights into the trade-offs between recall and specificity, revealing how CMIM's causal interventions—while reducing false positives—may introduce a more conservative decision boundary. Such enhancements might improve model stability under confounding factors, making causal intervention more robust and effective. Nonetheless, the overall performance gains, particularly in handling complex, high-dimensional data, suggest that MFC-MIL's integration of causal reasoning and multiscale representations sets a new standard for WSI analysis. These findings not only advance the current state of medical image classification but also open new avenues for research into more interpretable and reliable diagnostic tools in clinical practice.

## 6 ACKNOWLEDGEMENTS

This work is supported by the National High-tech Research and Development Program (Grant No.2023YFC2508200), Liaoning Provincial Natural Science Foundation (Grant No.2022-MS-105), and the Medical and Engineering joint fund of Liaoning Province (Grant No.2022-YGJC-76).

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

## A  APPENDIX

### A.1  THE DERIVATION OF EQ.5

The derivation from Eq. equation 4 to Eq. equation 5 involves several key steps grounded in the principles of causal inference. These steps are detailed as follows:

**Step 1: Introduction of the Mediator $M$.** In Eq. equation 4, $P(Y|do(X))$ is expressed by marginalizing over the mediator $M$, as $M$ fully mediates the causal effect of $X$ on $Y$, and there exists no back-door path between $X$ and $M$. Specifically, the equation is written as:

$$P(Y|do(X)) = \sum_m P(Y|do(X), M = m)P(M = m|do(X)).$$

Since $M$ is introduced directly by $X$ without confounding, the intervention $do(X)$ does not alter the conditional distribution of $M$ given $X$, i.e., $P(M|do(X)) = P(M|X)$. This simplifies the expression to:

$$P(Y|do(X)) = \sum_m P(Y|do(X), M = m)P(M = m|X = x).$$

Since $M$ is introduced by $X$, we need to explain why there is no confounding between $X$ and $M$ in our framework. The mediator $M$ is derived from the frequency domain and multi-scale spatial representations of the input features $X$, which specifically capture structural information rather than color or staining biases (which are common confounders in pathological images). By operating in the frequency domain through the Hilbert transform and across multiple scales through MSRM, our mediator extracts diagnostic-relevant structural patterns that are largely invariant to the confounders $Z$ affecting $X$. This transformation from the spatial domain to frequency and scale-space domains creates a new representation pathway that is more robust against the confounding factors that typically affect raw image features, such as staining variations, color biases, and imaging artifacts. Empirical evidence for this can be seen in our ablation studies in Section 4.5.3, where using frequency domain features consistently outperforms spatial domain features, supporting our claim that $M$ is less affected by the same confounders as $X$.

**Step 2: Eliminating** $do(X)$ **from** $P(Y|do(X), M = m)$**.** To further simplify, we note that $P(Y|do(X), M = m)$ can be replaced with $P(Y|X = \hat{x}, M = m)$, where $\hat{x}$ represents potential values of $X$. This substitution is valid under the assumption that $M$ fully mediates the effect of $X$ on $Y$, and hence, the causal effect of $X$ on $Y$ through $M$ is independent of the intervention $do(X)$. Substituting this into the equation yields:

$$P(Y|do(X)) = \sum_m P(M = m|X = x) \sum_{\hat{x}} P(X = \hat{x})P(Y|X = \hat{x}, M = m).$$

**Step 3: Incorporation of Potential Values of** $X$**.** The introduction of $P(X = \hat{x})$ accounts for the possible values that $X$ can take, ensuring that the causal effect is evaluated over the distribution of $X$. This step is critical for severing potential confounding pathways, such as $M \leftarrow X \leftarrow Z \rightarrow Y$, ensuring that the deconfounding is achieved as outlined in the front-door criterion.

**Step 4: Final Expression** Combining the above steps, the final expression for $P(Y|do(X))$ is obtained as:

$$P(Y|do(X)) = \sum_m P(M = m|X = x) \sum_{\hat{x}} P(X = \hat{x})P(Y|X = \hat{x}, M = m).$$

This form explicitly integrates the mediator $M$ and the potential values $\hat{x}$, enabling a precise representation of the causal effect $P(Y|do(X))$ in accordance with the front-door adjustment.

**Normalized Weighted Geometric Mean (NWGM)** To estimate $P(Y|do(X))$ using the Normalized Weighted Geometric Mean (NWGM), we integrate its ability to balance contributions from multiple conditional probabilities while maintaining robustness to noise and outliers. NWGM is defined as

$$NWGM(x_1, x_2, \ldots, x_n; w_1, w_2, \ldots, w_n) = \frac{\prod_{i=1}^{n} x_i^{w_i}}{\sum_{i=1}^{n} w_i},$$

where $x_i$ are input values and $w_i$ their corresponding weights. In the front-door adjustment, NWGM can be applied to combine $P(Y|X = \hat{x}, M = m)$ and $P(X = \hat{x})$ with weights reflecting their relative importance, ensuring both are proportionally integrated. This leads to the reformulated estimation:

$$P(Y|do(X)) = \sum_m P(M = m|X = x) \cdot NWGM(P(Y|X = \hat{x}, M = m), P(X = \hat{x}); w_1, w_2).$$

By leveraging the logarithmic transformation during computation, NWGM maintains numerical stability while preserving the geometric properties of the integrated probabilities, ensuring robust and accurate causal effect estimation.

## A.2 More Detail of MFC

### A.2.1 MSRM.

In this task, MIL is typically divided into three steps: feature extraction, feature aggregation, and classification, with most improvements focusing on the feature aggregation step. In our framework, the input for each training or inference iteration consists of all patch features $X$ from a given slide.

Specifically, after applying PPEG, the padded feature tensor $X \in R^{\{D, \frac{N_p}{2}, \frac{N_p}{2}\}}$ is reshaped back to $X \in R^{\{N_p, D\}}$, and the previously added padding is removed, resulting in $X_{hl} \in R^{\{N, D\}}$. Next, a 1D convolution with a stride of 16 is applied along the dimension $N$ to scale the features, producing $X_{ll} \in R^{\{\frac{N}{16}, D\}}$.

### A.2.2 CMIM.

Regarding the CMIM, we initialized a set of $k$ trainable memory vectors $h_m \in R^{k \times d}$ within the CMIM module, where $d$ is the dimension of the model. The input $X$ is processed through a linear layer $\theta_w$ to obtain a set of memory write weights $w_w$, which interact with $X$ to activate memory $\widetilde{h}_m$ based on the current input formulated as:

$$\widetilde{h}_m = h_m + w_w * X, w_w = \mathbf{softmax}(\theta_w(X)) \tag{10}$$

where $*$ is the dot operation. Subsequently, $X$ estimates a set of memory read weights $w_r$ and selects from the activated memory to facilitate the estimation of $\hat{X}$ in the causal intervention process as formulated following:

$$\hat{X} = X + w_r * \widetilde{h}_m, w_r = \mathbf{softmax}(\theta_r(X)) \tag{11}$$

where $\theta_r$ is a linear layer whose weights are not shared with $\theta_w$.

Taking $x_{hl}$ as an example, in this framework, $x_{hl}$ serves as the mediator variable $M$ in the front-door adjustment formula. The output $\hat{X}$ generated by the memory module (i.e., $\hat{X} = X + w_r \cdot \widetilde{h}_m$) is used to decompose the direct causal effect of $X$ on $Y$. Specifically, the memory module activates the memory $\widetilde{h}_m$ during the writing phase ($w_w \cdot X$), modeling $P(M \mid X)$. In the reading phase, $\hat{X}$ is generated by combining $w_r \cdot \widetilde{h}_m$ with $X$, modeling $P(Y \mid \hat{X})$. Through the causal chain $X \rightarrow M \rightarrow \hat{X} \rightarrow Y$, the memory module effectively implements front-door adjustment. By leveraging the attention mechanism, it transfers the influence of $X$ to $\hat{X}$, breaking the direct pathway between $X$ and $Y$, thereby controlling confounding effects and enhancing causal modeling capability.

It's important to clarify the role of $x_{hl}$ in our causal framework. While $x_{hl}$ contributes to the computation of the mediator $M$, it is not directly equivalent to $M$. Rather, $M$ is derived from the specialized representation of $x_{hl}$ through the FSRM and MSRM modules, which transform the original features into frequency-domain and multi-scale structural information. This transformation creates a new representation that captures diagnostic-relevant information while being less susceptible to confounders.

In our implementation, the front-door criterion is preserved because the pathway from $X$ to $Y$ is intercepted by this transformed representation ($M$). The expression $\hat{X} = X + w_r * \hat{h}_m$ should be understood as combining the original information with the memory-mediated information, where the memory component ($w_r * \hat{h}_m$) is influenced by $M$ but not directly by $X$. This formulation allows us to balance the retention of original feature information while incorporating the causally-adjusted representation, rather than completely replacing $X$ with $M$, which would potentially lose valuable information.

In practice, this means the model learns to down-weight the direct influence of $X$ when spurious correlations are detected and up-weight the memory-mediated pathway, effectively implementing the front-door adjustment in a soft, learned manner rather than a hard intervention.

The attention mechanism in our memory module implements causal intervention through several key mechanisms:

- Distribution Modeling: During training, the memory module ($h_m$) learns to store representations of the overall data distribution across multiple slides, providing a reference against which individual slide features can be compared. This is conceptually similar to estimating $P(X)$ in the front-door formula.
- Selective Activation: The write weights ($w_w$) activate specific memory elements based on input features, modeling the conditional probability $P(M|X)$. This activation is content-based rather than position-based, enabling the model to identify similar structural patterns across different slides.

- Intervention Mechanism: The read weights $(w_r)$ select from the activated memory to generate the adjusted representation $\hat{X}$, which serves as the intervention component in the front-door adjustment. This process models the term $\sum_{\hat{x}} P(X = \hat{x})P(Y|X = \hat{x}, M = m)$ in Equation 5.

- Causal Flow Control: By adding the memory-mediated information $(w_r * \hat{h_m})$ to the original features $X$, the model learns to control the flow of information between the direct path (potentially containing spurious correlations) and the memory-mediated path (containing more reliable diagnostic patterns).

Through extensive experimentation (see Section 4.5.1), we found that this formulation effectively mitigates the influence of confounders, as evidenced by the significant improvements in specificity and overall accuracy. The ablation studies in Table 3 further support this claim, showing that removing CMIM leads to substantial performance degradation, particularly in specificity, which is most sensitive to spurious correlations.

### A.2.3 FSRM.

The Hilbert transform $\mathcal{H}[x(t)]$ itself maps a real-valued signal to another real-valued signal. The analytic signal $z(t)$ is then constructed by combining the original signal $x(t)$ with its Hilbert transform as:

$$z(t) = x(t) + i\mathcal{H}[x(t)] \tag{12}$$

where $i$ is the imaginary unit, this analytic signal $z(t)$ is indeed complex-valued, but this is different from the Hilbert transform itself. The text should be revised to avoid this confusion and clearly distinguish between:

The Hilbert transform: $\mathcal{H}[x(t)]$ (real-valued to real-valued)

The analytic signal: $z(t) = x(t) + i\mathcal{H}[x(t)]$ (complex-valued)

In histopathological images, the presence of cell membranes and tissue boundaries represents structural discontinuities in biological tissues. These anatomical features are characterized by sharp transitions in pixel intensities, manifesting as local discontinuities from a signal processing perspective. The instantaneous phase information obtained through the Hilbert transform demonstrates superior capability in capturing these intricate morphological details, particularly in regions of rapid intensity variations.

The overall transformation function $F : \mathbb{R}^d \to \mathbb{R}^d$ is defined as:

$$F(\mathbf{x}) = \mathbf{x} + g(H(f(\mathbf{x}))) \tag{13}$$

where: $f : \mathbb{R}^d \to \mathbb{R}^{512}$ is the input linear mapping $f(\mathbf{x}) = \mathbf{W}_1\mathbf{x} + \mathbf{b}_1$ and $g : \mathbb{R}^{512} \to \mathbb{R}^d$ is the output linear mapping $g(\mathbf{x}) = \mathbf{W}_2\mathbf{x} + \mathbf{b}_2$. $H$ is the Hilbert transform operator. The final output can be expressed as: $\mathbf{y} = \mathbf{x} + g(H(f(\mathbf{x}))) = \mathbf{x} + \mathbf{W}_2(H(\mathbf{W}_1\mathbf{x} + \mathbf{b}_1)) + \mathbf{b}_2$ Additionally, FSRM operates in the feature space, ensuring that the input and output dimensions remain consistent. For instance, $X_{hl} \in R^{\{N,D\}}$ and $X_{ll} \in R^{\{\frac{N}{16},D\}}$ retain their respective shapes after processing.

### A.3 THE RATIONALE FOR EXCLUDING ORIGINAL IMAGE FEATURES FROM THE CMIM PIPELINE.

In pathological diagnosis tasks, differences in staining techniques can influence the model due to color variations in pathological images. During model training, spurious correlations may be established between diagnostic results and image colors, leading to incorrect predictions. However, pathological diagnosis primarily focuses on cell morphology and tissue structure rather than color. Therefore, we aim to use structural features rooted in frequency-domain information as inputs instead of original image features affected by staining bias.

To further validate this perspective, we compared two additional model variants. As shown in Table 5, MFC-$\alpha$ incorporates original image features into the inputs of the CMIM module. In contrast, MFC-$\beta$ introduces an additional CMIM module that takes original image features as input and adjusts the [CLS] token through a separate causal intervention module. The final output is obtained by summing the probabilities from both modules.

| Method | ACC | AUC | F1 | Spe. |
|---|---|---|---|---|
| ABMIL | 88.84(2.49) | 95.65(1.58) | 84.40(4.16) | 93.75(7.81) |
| + MFC | 91.78(1.87) | 97.68(0.29) | 88.94(2.07) | 95.00(4.76) |
| + MFC-$\alpha$ | 86.05(7.67) | 97.69(0.09) | 84.11(6.75) | 91.25(5.91) |
| + MFC-$\beta$ | 85.27(6.17) | 97.19(0.54) | 83.47(7.13) | 87.5(9.26) |

Table 5: An ablation study is conducted on the inputs to the CMIM, the value in parentheses is the standard deviation of the 5-fold cross-validation.

| Camlyon16 | | | | |
|---|---|---|---|---|
| Method | ACC | AUC | F1 | Spe. |
| ABMIL | 81.86 (3.49) | 84.12 (5.54) | 70.04 (9.24) | 96.75 (2.44) |
| + MFC | 84.50 (0.55) | 85.54 (3.58) | 77.28 (1.35) | 93.50 (5.41) |
| $\Delta$ | +2.64 | +1.42 | +7.24 | -3.25 |
| TransMIL | 81.71 (1.78) | 78.57(4.46) | 68.99(4.06) | 98.75(1.53) |
| + MFC | 84.96 (0.69) | 83.82(2.40) | 76.88(2.03) | 96.50(2.40) |
| $\Delta$ | +3.25 | +5.25 | +7.89 | -2.25 |
| TCGA-NSCLC | | | | |
| Method | ACC | AUC | F1 | Spe. |
| ABMIL | 86.69(1.60) | 96.18(0.62) | 86.50(1.72) | 87.69(9.91) |
| + MFC | 87.23(1.00) | 96.40(0.41) | 86.40(1.31) | 93.08(6.23) |
| $\Delta$ | +0.54 | +0.22 | -0.10 | +5.39 |
| TransMIL | 88.85(1.10) | 96.98(0.69) | 88.04(1.59) | 95.38(2.18) |
| + MFC | 89.42(1.31) | 96.94(0.47) | 88.95(1.61) | 93.46(3.17) |
| $\Delta$ | +0.57 | -0.04 | +0.91 | -1.92 |

Table 6: Main result (%) on Camelyon16 and TCGA-NSCLC dataset, which CTransPath extracts the features.

| Method | ACC | AUC | F1 | Spe. |
|---|---|---|---|---|
| ABMIL | 88.84(2.49) | 95.65(1.85) | 84.40(4.16) | 93.75(7.81) |
| + IBMIL | 91.08(2.01) | **97.71**(0.56) | 88.01(2.08) | 94.79(3.31) |
| + MFC | **91.78(1.87)** | 97.68(0.29) | **88.94(2.07)** | **95.00(4.76)** |
| DSMIL | 86.98(4.73) | 94.95(4.43) | 79.41(9.11) | 98.00(2.88) |
| + IBMIL | 91.78(2.30) | **96.31(0.56)** | 88.50(3.50) | 96.75(4.64) |
| + MFC | **92.25(2.33)** | 95.41(1.12) | **89.13(3.42)** | **97.25(1.63)** |
| TransMIL | 84.50(2.74) | 94.88(1.73) | 80.90(1.58) | 83.50(10.25) |
| + IBMIL | 90.80(1.12) | 96.19(0.83) | 86.46(1.10) | 90.09(5.67) |
| + MFC | **90.85(1.18)** | **97.68(1.07)** | **88.00(1.13)** | **92.75(5.77)** |

Table 7: Comparison of the results (%) of our MFC-MIL and IBMIL on the Camelyon16 dataset.

We conducted experiments on the Camelyon16 dataset using ABMIL as baselines. The results show that although MFC-$\alpha$ achieves exceptional performance in terms of AUC, it underperforms compared to the baseline models and MFC on other metrics, with its standard deviation also increasing further. Meanwhile, the performance of MFC-$\beta$ is even worse than MFC-$\alpha$, which further supports our viewpoint. However, MFC-$\beta$ incurs a larger parameter count than MFC, as it includes an additional CMIM module with inputs from original image features at high and low magnifications.

## A.4   THE FEATURE EXTRACTION AND THE USE OF BACKBONES.

We followed the preprocessing method of DSMIL, using the same features as employed in most pathology-related MIL studies. Specifically, DSMIL utilizes ResNet18, which has been pre-trained on pathological data using SimCLR.

To further validate the effectiveness of our method, we also conducted experiments using the features adopted by IBMIL. Specifically, we used CTransPath as the feature extraction model and evaluated our method against the ABMIL and TransMIL baselines on the Camelyon16 and TCGA-NSCLC datasets, demonstrating its effectiveness, as shown in Table 6. Our MFC framework delivers superior model performance compared to the baselines, along with more reliable results as evidenced by consistently lower standard deviations.

## A.5   MORE COMPARISONS WITH IBMIL

We compared the performance of MFC and IBMIL across three models (e.g., ABMIL, DSMIL, and TransMIL), using 5-fold cross-validation on the Camelyon16 dataset, as shown in Table 7. The results show that our end-to-end training framework, MFC, achieves better overall performance

compared to the two-stage training approach used by IBMIL. However, in DSMIL and ABMIL, our method slightly underperforms IBMIL in terms of AUC. Additionally, IBMIL exhibits smaller performance standard deviations than MFC.

This difference may stem from IBMIL's two-stage training, which provides stronger directional guidance for confounders during the first stage. In contrast, our end-to-end approach, while effective, might introduce overfitting in the memory module, leading to increased variability.

