# OpenReview forum: "A Multiscale Frequency Domain Causal Framework for Enhanced Pathological Analysis"
_ICLR.cc/2025/Conference — ICLR 2025 Poster_

### Official Review · Reviewer_PUSw · 2024-10-31

**Soundness:** 4
**Presentation:** 4
**Contribution:** 4
**Rating:** 8
**Confidence:** 5

**Summary:**

This paper proposes a multi-scale frequency-domain causal framework (MFC-MIL) for the classification of pathological images. The paper addresses the limitations of multiple instance learning (MIL) in whole-slide image (WSI) pathology analysis by identifying and tackling issues related to data bias and unobservable confounding variables. By incorporating causal intervention and multi-scale feature representation, the model demonstrates significant performance improvements across various datasets. Overall, the methodology is well-designed, demonstrates strong innovation, and is thoroughly validated. The MFC-MIL framework provides substantial practical and theoretical contributions, effectively enhancing the accuracy and robustness of pathological image classification.

**Strengths:**

1. The MFC framework is novel, combining multi-scale spatial and frequency-domain feature representations with a causal memory intervention module to address data bias in pathology image analysis. This multi-module, collaborative approach to causal intervention is particularly innovative.

2. The MSRM module integrates information at both low and high magnifications, with multi-scale convolutions and positional encoding capturing tissue structures at various levels. The FSRM module effectively captures frequency-domain information through Hilbert transforms, introducing new structural features that improve the model's discrimination capability with complex data.

3. The effectiveness of the MFC framework is validated on the Camelyon16 and TCGA-NSCLC datasets. Through comprehensive comparisons, ablation studies, and parameter investigations, the model’s performance is thoroughly evaluated, and the results are convincing.

4. The proposed MFC framework surpasses existing methods in metrics such as accuracy and F1 score and also enhances model interpretability. The causal intervention module (CMIM) reduces the influence of non-causal features through a memory selection mechanism, offering a novel theoretical perspective for pathology analysis.

**Weaknesses:**

1. While the MSRM and FSRM modules demonstrate strong performance in experiments, further explanations regarding the theoretical motivations and details of each module would enhance clarity. For example, connecting the role of the Hilbert transform in frequency-domain feature extraction to specific pathology image characteristics would improve readers’ understanding of its biological significance.

2. Although the ablation studies validate the effectiveness of each module, the analysis could delve deeper into how individual modules impact various evaluation metrics (e.g., AUC, F1 score). Adding a discussion on the underlying reasons behind the observed experimental phenomena could enhance the depth of the experimental results.

3. The addition of the FSRM module improves classification performance, but the comparison with traditional frequency-domain methods (e.g., FFT) is somewhat brief. Providing a more detailed comparative analysis of different frequency-domain feature extraction techniques on pathology images could further highlight the practical effectiveness of the FSRM module.

**Questions:**

1. Does the memory selection mechanism in the CMIM module risk overfitting? The variation in the number of memory slots within the CMIM seems to significantly impact performance. Does this imply a potential risk of overfitting with the memory selection mechanism? Is this mechanism equally effective across other datasets or different types of pathology images?

2. The paper presents two feature representation modules that enhance the model’s memory capabilities. The authors may consider referring to the following two papers for future work: *Unsupervised Multi-Domain Progressive Stain Transfer Guided by Style Encoding Dictionary* and *Wavelet Encoding Network for Inertial Signal Enhancement via Feature Supervision.* Both studies employ a regularization method based on Rényi entropy, which significantly enhances model representation and memory capacity. The authors could consider introducing this approach in future work. Such a design might improve the distinctiveness and information density of features, thereby increasing model stability under different confounding factors, minimizing interference from non-causal features, and making causal intervention more effective.
This suggestion is just an academic discussion and does not imply that the authors need to supplement any additional experiments. The sole purpose is to inspire the authors’ future work and model design. Adding a section on “Future Work” could further elevate the significance and impact of this paper.

---

### Official Review · Reviewer_3jEd · 2024-11-03

**Soundness:** 2
**Presentation:** 1
**Contribution:** 2
**Rating:** 3
**Confidence:** 4

**Summary:**

This paper presents a Multi-Scale Frequency Domain Causal (**MFC**) framework for Multiple Instance Learning (MIL) applied to histopathology Whole Slide Imaging (WSI) which aims to improve both accuracy and generalisation by addressing three areas: data bias from unobservable confounders, integration of features across multiple scales, and interference from varying staining techniques. There are three main components, designed to address these issues: Causal Memory Intervention Module (**CMIM**), Multiscale Spatial Representation Module (**MSRM**) and Frequency-domain Structural Representation Module (**FSRM**). MFC is designed to be used on top of other MIL methods. The pipeline seems to consist of the following steps:

- the MSRM takes as input N patches coming from WSI, pads them to an even number and reshapes them into a square, before passing this into three Conv2D operations (kernel 7, 3, 5) before unpadding, reshaping and keeping this output as a high resolution feature vector $X_{hl}$. $X_{hl}$ is then passed through 3 parallel Conv1D operation (kernel 16, dilation 1, 3, 5) to obtain N/16 low resolution feature vectors $X_{hl}$. These feature vectors are used as downstream input to FSRM.

- The analytical signal of the above mentioned feature vectors is extracted by using the Hilbert Transform in parallel FSRMs. The processed features are mapped back to the original input space via a projection layer.

- These features are input into parallel CMIMs, which initialise a set of trainable parameters with length k, which are then combined with attention-weighted inputs to select relevant memory elements and classified.

The authors implement their framework on top of 5 well-know MIL models and two benchmark datasets (CAMELYON16 and TCGA-NSCLC) and report an increase in performance across base model and datasets. They also compare with another causal framework IBMIL and perform ablation on the three proposed components.

**Strengths:**

This paper presents some interesting and potentially valuable ideas for the field of computation pathology. The authors correctly identify three challenging areas: the potential presence of unobservable confounders in the data, the need to integrate local and global scales, and the noise introduced through variance in staining techniques. Their proposed framework attempts to address these challenges through a combination of targeted techniques: using a memory-based intervention for deconfounding, a multi-scale spatial representations for capturing both cellular and tissue-level features, and frequency domain analysis for reducing effects from variance in staining. The empirical results show consistent improvements across multiple baseline models and two different datasets, suggesting the potential value of this approach. The framework is designed to be modular and can be integrated with existing MIL architectures, which makes could make it useful to improve upon already existing approaches.

**Weaknesses:**

Unfortunately, I find this paper reads as if it wasn't properly finished. The explanation of the methods in general is unclear and disorganised, and in my opinion doesn't properly motivate the design choices. There's a lot of typos and inaccuracies and fundamentally, there's nothing to really back up the claims the authors are making which makes it hard to assess the strength/relevance of their contributions.

**MSRM**:

- How do you format the input coming from the MIL model? You say you perform MSRM as a first step. It requires N feature vector corresponding to image patches, so I assume this isn't from the output of the models? The Figure 1 implies it is, so this either needs to be amended or explained more clearly.

- The padding strategy is not explained: you say "X is padded in the PPEG" but don't say how. You don't specify if/how you pad for the convolutions. You then say "padding is removed, and the original dimensions are restored", which doesn't explain how this is possible given the convolutions. The 1D convolution part is also unclear to me, as I don't understand how you apply the 1D convs to the output of 2D convolutions.

- I also feel there are discrepancies between the visual illustration and the textual description. For example in Figure 2 (b) you graphically illustrate the MSRM module, but it seems to contradict the textual description: you just show a PPEG block with no details of internal operations, which splits into MaxPooling and three Conv1D, with additional GeLU and Linear layer.

Overall, the description is confusing and lacks technical detail which would allow the reader to more easily understand your approach and why it makes sense.

**CMIM**:

I read this section and was left wondering why this is a causal model with front-door intervention? First off, the paper skips the steps showing how the do-operator terms are manipulated to get from Eq. 4 to 5. It states the derivation is in the Appendix, but there is no Appendix...

Furthermore, assuming eq. 5, why does it justify "utilizing a memory module to estimate the overall distribution of the dataset during training and to refine the estimation of $\hat{x}$ through attention-based sampling."? You then say this "selected memory is further sampled and used as $\hat{x}$ in the front-door intervention as illustrated in Figure 2 (a). Finally, we employ the Normalized Weighted Geometric Mean [...] to estimate the equation."

I don't understand how this description or the illustration in Figure 2 (a) relates to a do-operator or front-door intervention. Selection via attention doesn't obviously implement the do-operator. Furthermore, how does a learned memory parameter acts as a mediator here? And how does this learned memory act as a front-door intervention?

It would be great if you could backup your claims more fully. This is one of the central aspects of this paper, so it needs to be carefully explained and argued.

**FSRM**:

- Line 234: "By applying the Hilbert transform to a signal $x(t)$, we obtain a complex-valued function $x_a(t)$, where the original signal forms the real part, and the Hilbert transform provides the imaginary part."

My understanding is the Hilbert Transform takes a function of a real variable and produces another function of a real variable. Rather, here you obtain a complex-valued function by multiplying $\hat{x}(t)$ by the imaginary unit $j$, which is you analytic signal $x_{a}(t)$. I think you're confusing the definition of Hilbert Transform with that of the analytical signal?

- Line 264 - 268: "The core of the module is the Hilbert transform, which extracts the analytic signal of the features, providing a comprehensive representation of both magnitude and phase information. An optional phase extraction step can be employed to focus specifically on the phase components, which often carry significant structural information."

Figure 2 (c) seems to imply you apply FSRM in the RGB domain, but from the text above and Figure 1, it would seem this is applied in the feature space domain. In general, more clarity on how you're applying the analytical signal to your input would be appreciated. What are the input, output sizes? How exactly is the Hilbert transform applied to the feature vectors? How is the phase extraction used? Also, explaining how the analytical signal is being used as a filter on the images and what type of feature it can extract would also be helpful.

**Results**:

- Results are shown comparing this framework added on top of other MIL methods, but really this should be compared to other frameworks which also aim to reduce spurious correlations or increase generalisation. For example, you only compare (TransMIL + IBMIL) to (TransMIL + MFC), but you should compare IBMIL vs MFC in all cases.

- There are no uncertainty measures on the results, so you can't make any claims as to the significance of the results. At a minimum you should be including standard deviation.

- Line 404: you have no backing to the claim CMIM modules is more effective at capturing causal features. Why is it better? How can you illustrate this?

- Line 429: again, you're not showing the MSRM module is actually effectively picking up information from different scales.

- Line 456: FFT, DCT, and DWT are not even defined in the text.

**Further comments**:

- Line 211 - 213: "After implementing CMIM, we further refined the estimation of the mediator, particularly by integrating low-magnification tissue information with high-magnification cellular information, which are both crucial in the diagnostic process."

Do you implement CMIM, then MSRM as implied here - or MSRM first as implied in Figure 1.

- Line 254: you're suddenly calling your FSRM module "fdsr".
- Line 279: Metric
- Line 302: MODELS
- Line 308: CLAM isn't defined properly.
- Line 316: the design of your train/test split should go in implementation details.
- Line 320: Moreover,
- Line 324: Camelyon16

**Questions:**

Based on the comments I have expanded upon above, here is a list of questions I believe the authors need to address.

**MSRM**:

- How exactly is the input from MIL models formatted for your framework?
- What is the precise padding strategy in PPEG? Please provide implementation details.
- How do you maintain spatial dimensions through the Conv2D operations?
- How are 1D convolutions applied to the output of 2D convolutions?
- Could you provide a detailed diagram showing the internal operations of PPEG?
- What are the exact dimensions at each step of the MSRM pipeline?

**CMIM**:

- Could you provide the missing derivation showing how you get from Eq. 4 to 5?
- How does selecting memory elements through attention implement the do-operator?
- How do learned memory parameters act as mediators in your framework?
- Could you mathematically justify why your memory-based approach implements front-door intervention?
- How exactly is NWGM used to estimate the final equation?
- What is the exact implementation of your attention mechanism?

**FSRM**:

- Is FSRM applied in RGB domain or feature space?
- What are the exact input and output dimensions?
- How exactly is the Hilbert transform applied to feature vectors?
- How is the phase extraction implemented and used?
- Could you explain how the analytic signal acts as a filter and what features it extracts?
- Could you clarify the distinction between Hilbert Transform and analytic signal in your implementation?

**Results**:

- Could you provide uncertainty measures (e.g., standard deviation) for your results?
- Why not compare IBMIL vs MFC across all baseline models?
- How do you measure and validate that CMIM captures causal features?
- How do you verify MSRM effectively captures information at different scales?

**Framework**:

- What is the correct order of operations - CMIM then MSRM, or MSRM first?
- Could you provide a consistent diagram showing the exact flow of information?

---

### Official Review · Reviewer_7Fj5 · 2024-11-04

**Soundness:** 3
**Presentation:** 4
**Contribution:** 3
**Rating:** 8
**Confidence:** 5

**Summary:**

The authors stated that the existing DL method learns from a single magnification of pathological images, which limits the model’s accuracy. The authors proposed a new model named MFC, which includes the CMIM, MSRM, and FSRM modules. CMIM is designed to address the issue of spurious correlations by preserving diagnostic features as learnable memory elements and facilitating causal interventions. It reduces the influence of confounders, ensuring that the model makes decisions based on causal relationships rather than coincidental patterns in the data. This module helps improve model robustness by using attention-weighted sampling to refine the estimation of memory elements that contribute to more accurate predictions. MSRM integrates information across multiple scales to capture spatial relationships between different levels of image detail, such as tissue structures at low magnification and cellular details at high magnification. This module applies position-aware patch embedding and convolutions with various kernel sizes to extract features with different receptive fields, enhancing the model’s ability to process multilevel information and improving its representation capabilities. FSRM leverages the Hilbert transform to analyze the frequency domain of the image features, capturing both amplitude and phase information. This module helps identify subtle textural and structural variations that might be overlooked in spatial analyses, making it especially valuable for reducing the influence of staining techniques and color variations in pathology images. By incorporating frequency information, FSRM strengthens the model's ability to extract diagnostic features more effectively.

**Strengths:**

The paper is well written and very clear. It is very easy to follow.

**Weaknesses:**

The reason for using ResNet18 as the feature extractor is not clear. The authors did not mention whether they tested different models.

**Questions:**

1.	The font in the figure is somewhat small.
2.	The caption for Figure 2 needs additional explanation.
3.	In line 200, “Where” should be in lowercase.
4.	The title "3.3.2 METHODS" should be changed to avoid confusing the audience.
5.	Will different backbone feature extraction models impact the performance of the proposed method? Why was ResNet18 chosen?
6.	Why was a mini-batch size of 1 selected?
7.	The data preprocessing steps and the choice of feature extraction model are not clear, considering that the proposed method depends on the extracted features.

---

### Official Review · Reviewer_p1Pm · 2024-11-08

**Soundness:** 2
**Presentation:** 2
**Contribution:** 2
**Rating:** 5
**Confidence:** 5

**Summary:**

The paper proposes a multi-scale frequency domain causal framework for WSI classification that addresses the challenges of data bias and unobservable confounders. The framework comprises three key components: 1) a multi-layer spatial representation module, 2) a frequency domain structure representation module, and 3) a causal memory intervention module. The authors present comprehensive experimental results to validate their approach.

**Strengths:**

1.	The extensive experimental validation effectively demonstrates the performance of both individual components and the complete system
2.	The novel application of frequency domain analysis to mitigate irrelevant factors such as color variations represents an innovative approach
3.	The overall paper structure and flow are well-organized, though some core concepts would benefit from more precise articulation

**Weaknesses:**

1.	There are significant inconsistencies between the abstract and main content. The abstract references concepts such as information bottleneck and backbone fine-tuning that are absent from the paper. While the general writing is clear, the technical descriptions of the proposed modules lack sufficient precision and detail
2.	The mathematical formulation of the novel components is inadequate. While the paper includes basic equations from prior work, the core innovative concepts lack rigorous description. Specific areas requiring mathematical formalization include: 1) Lines 200-207: Memory module operations and 2) Lines 262-269: Feature transformation procedures
3.	The causal inference framework appears to be limited to Hilbert-transformed images, excluding original image data. This design choice requires either theoretical justification or empirical validation
4.	Several claims made in the conclusion lack supporting evidence in the main text, including: 1) Trade-offs between recall and specificity, and 2) CMIM's capability to reduce false positives
5.	Minor Issues: a. Inconsistent capitalization (e.g., "pathology" in abstract, "FDSR" in line 254) b. Strange visualization in Figure 2(b) c. Absence of proper introduction for technical abbreviations (FFT, DCT)

**Questions:**

1.	What is the rationale for excluding original image features from the CMIM pipeline?
2.	Please provide detailed specifications of the modules:
      a.	Memory elements
      b.	Attention mechanism implementation
      c.	Memory element selection criteria
      d.	Weighting methodology
3.	Please clarify the technical implementation details:
      a.	Hilbert transform application to feature vectors
      b.	Projection layer architecture
      c.	Scope and implementation of residual connections (FSRM module vs. complete MFC)

---

### Meta-Review · Area_Chair_scc7 · 2024-12-16

**Metareview:**

This paper proposes a plug-and-play model for enhancing whole slide image analysis by introducing three modules, i.e., the Multi-scale Spatial Representation Module (MSRM), the Frequency Domain Structure Representation Module (FSRM), and the Causal Memory Intervention Module (CMIM). The proposed framework can be applied to existing MIL methods to further boost diagnosis performance. Comprehensive experiments have shown promising classification results for WSI analysis.

This paper received mixed review ratings, including 2x strong accept, 1x marginally below the acceptance threshold, and 1x reject. The reviewers' questions regarding this work centered around paper writing, methodology design, and requirements for detailed explanations of certain aspects. The authors have addressed most concerns during the discussion.

Given the merit of novelty and comprehensiveness of this work, I recommend acceptance. In the meantime, I strongly suggest that the authors should make further improvements based on reviewer p1Pm's and reviewer 3jEd's comments.

**Additional Comments On Reviewer Discussion:**

Reviewers' concerns mainly focus on insufficient illustration and explanation of the proposed methodology. Although the authors have managed to solve most of them, I strongly recommend a further modification for final version of the paper.

---

### Decision · Program_Chairs · 2025-01-22

Accept (Poster)